# Suicidal Ideation and Substance Use Among Middle and High School Students in Morocco

**DOI:** 10.3390/healthcare13101178

**Published:** 2025-05-19

**Authors:** Abdelmounaim Baslam, Hajar Azraida, Samia Boussaa, Abderrahman Chait

**Affiliations:** 1Laboratory of Pharmacology, Neurobiology, Anthropobiology and Environment, Faculty of Sciences Semlalia, Cadi Ayyad University, Marrakech 40000, Morocco; 2ISPITS–Higher Institute of Nursing and Health Techniques, Ministry of Health and Social Protection, Rabat 10020, Morocco

**Keywords:** substance use, suicidal ideation, depression, psychoactive substances, student, Morocco

## Abstract

**Background/Objectives**: Suicide is a major public health concern with severe consequences for individuals, families, and communities. Each year, approximately 800,000 individuals die by suicide, equating to one suicide-related death every 40 s. This study aims to determine the prevalence of suicidal ideation and psychoactive substance use among middle and high school students in Morocco and to examine the relationship between suicidal ideation, substance use, depression, and early trauma. **Methods**: From January to June 2023, a cross-sectional study was conducted in the Marrakech region of central Morocco among middle and high school students selected using stratified cluster random sampling. Data were collected through anonymous self-administered questionnaires. The Suicidal Ideation Attributes Scale (SIDAS) was used to assess the frequency of suicidal ideation among participants. The Diagnostic and Statistical Manual of Mental Disorders (DSM) criteria were applied for substance use and depression assessment, while early trauma was evaluated using the Adverse Childhood Experiences (ACE) questionnaire. **Results**: A total of 791 students participated in the study. The prevalence of suicidal ideation was 16.66%, while 25% of students reported current psychoactive substance use. Significant positive correlations were found between suicidal ideation and early trauma (*r* = 0.12; *p* < 0.004), depression (*r* = 0.52; *p* < 0.001), and substance use (*r* = 0.12; *p* < 0.001). **Conclusions**: The findings highlight the importance of considering multiple risk factors in suicide assessment and prevention. The interplay between these factors often exhibits bidirectional and significant associations. Implementing early screening, management, and treatment strategies for individuals exhibiting suicidal ideation and mental health disorders is crucial to reducing the burden of suicide.

## 1. Introduction

Suicide is a major public health concern, with severe consequences for individuals, families, and communities. Each year, approximately 800,000 lives are lost to suicide, equating to one suicide-related death every 40 s [1]. Suicide accounts for 1.4% of global deaths, making it the 15th leading cause of mortality worldwide and the 2nd leading cause of death among individuals aged 15 to 29 [2]. It is influenced by a range of risk factors, including psychiatric disorders, substance abuse, psychological distress, genetic predisposition, and cultural and societal contexts [3].

Suicidal ideation refers to thoughts, contemplation, or planning related to suicide. It is often a precursor to suicide attempts or suicide itself, reflecting a tendency to view death as a solution to seemingly insurmountable life challenges [4]. According to Reynolds, suicidal ideation manifests as thoughts, ideas, or plans that may lead to self-harm or death, becoming more severe when individuals are unable to overcome their difficulties [5].

The mental well-being of adolescents and young adults is of paramount importance due to the impact of mental disorders on their present and future lives, as well as on subsequent generations. The challenges faced by young people can disrupt their cognitive, emotional, and behavioral patterns [6]. Several factors contribute to suicidal ideation among youth, whether they are students or young professionals. These include psychoactive substance use, academic difficulties, psychological disorders, poor social and familial relationships, aggression or impulsivity, and engagement in risky sexual behaviors, among others. Collectively, these factors can lead to feelings of despair and an increased risk of suicidal ideation. Recognizing and addressing these underlying issues is crucial for preventing the onset and persistence of suicidal thoughts among young people [7,8,9].

Research conducted across the Middle East and North Africa has revealed significant associations between suicidal ideation and substance use among youth, albeit with limited studies focusing specifically on these issues [10]. The scarcity of research in this region is exacerbated by cultural and social barriers, such as underreporting driven by social stigma and cultural constraints, leading to a gap in understanding the depth of these issues [11]. In particular, studies have shown that substance use, including alcohol and tobacco, is closely linked with suicidal ideation and attempts, yet much of the existing literature focuses on broader mental health conditions rather than substance use specifically. Addressing this gap could help better inform intervention strategies aimed at reducing suicide rates among youth in the region [12].

Earlier studies indicate that hopelessness is associated with suicide and can predict suicidal tendencies [13,14]. However, its predictive value diminishes when depression is taken into account, suggesting that the link between hopelessness and suicidal behaviors may depend on specific conditions [15]. In this regard, depression is a key factor in the development of suicidal thoughts among young individuals and is most frequently observed in adolescents who have attempted suicide. Research has demonstrated a positive correlation between stress, anxiety, depression, and the occurrence of suicidal thoughts [16,17].

Approximately 79% of adolescent suicides occur in low- and middle-income countries, which often lack comprehensive research on suicide, its associated risk factors, and effective prevention strategies [18]. Notably, the Middle East and North Africa have seen relatively limited research due to various factors, such as underreporting of cases driven by social stigma and cultural constraints [19]. According to the 2014 World Health Organization (WHO) report on suicide, Morocco recorded the highest suicide rate among Arab nations. Between 2000 and 2012, the incidence of suicide in Morocco surged by 97%, rising from 2.7 to 5.3 per 100,000 inhabitants [20]. A study conducted in 2017 on suicide in Morocco found that 16.6% of adolescents had experienced suicidal ideation. The study also identified a positive correlation between suicide rates and factors such as age, food insecurity, anxiety, feelings of isolation, bullying, and substance abuse [21].

Morocco’s socio-economic context is unique and plays a pivotal role in the well-being of its youth. The economic challenges faced by the younger generation, coupled with a rapidly changing cultural landscape, influence their sense of hope and future prospects. Traditional family structures, coupled with modern pressures, can exacerbate feelings of isolation and hopelessness among adolescents [22]. These cultural dynamics, along with increasing exposure to substance use, make Moroccan youth particularly vulnerable to the development of suicidal ideation.

The role of family socialization in Morocco further complicates these dynamics. While traditional family values remain significant, modern pressures—such as academic expectations, financial strain, and social media influence—have led to a shift in family interaction patterns, potentially contributing to an increased sense of isolation among adolescents [11].

This study focuses on adolescents and young adults, specifically in the Moroccan context, which is marked by unique cultural and socio-economic factors that influence mental well-being. The age range of the subjects, typically between 14 and 18 years, is a critical developmental period during which the risk for suicidal ideation and substance use may be heightened.

While suicidal ideation, substance use, and early trauma are common issues among students, research on their prevalence and associated risk factors remains scarce. This cross-sectional study aims to determine the prevalence of suicidal ideation and psychoactive substance use among middle and high school students in Morocco and to examine the relationship between suicidal ideation, substance use, depression, and early trauma.

## 2. Materials and Methods

### 2.1. Study Design and Setting

This cross-sectional study was conducted between January 2023 and June 2023, encompassing a total of 791 participants who were administered a standardized structured questionnaire. The questionnaire included sections assessing sociodemographic characteristics, substance use behaviors, suicidal ideation, depressive symptoms, and early trauma exposure. It incorporated validated scales such as the Ask Suicide-Screening Questions (ASQ) for suicide risk, the Suicidal Ideation Attributes Scale (SIDAS) for suicidal ideation frequency, the Adverse Childhood Experiences (ACE) questionnaire for early trauma assessment, and the DSM-5 criteria for substance use disorders and depression. To meet the inclusion criteria, participants had to be enrolled in middle or high school during the 2022–2023 academic year in various educational institutions across the Marrakech region, central Morocco.

### 2.2. Participants and Recruitment

A stratified cluster random sampling technique was employed to ensure the representativeness of the results across the entire target population, with all selected individuals having an equal chance of participation. In the first stage, to achieve a geographically representative sample, we allocated participants evenly, selecting 100 students from each of the five urban zones of Marrakech, along with three rural zones. In the second stage, schools were proportionally divided between high schools and middle schools. In the third stage, two schools were randomly chosen from each zone, including one high school and one middle school. Finally, individuals meeting the predefined inclusion criteria were randomly selected to participate in the study.

The sample size was calculated using the following formula [23]:$$n=\frac{\frac{Z^{2}\times P\times \left(1-P\right)}{e^{2}}}{1+(\frac{Z^{2}\times P\times \left(1-P\right)}{e^{2}N})}=\frac{\frac{{\left(1.96\right)}^{2}\times 0.5\times \left(1-0.5\right)}{{\left(0.05\right)}^{2}}}{1+(\frac{{\left(1.96\right)}^{2}\times 0.5\times \left(1-0.5\right)}{{\left(0.05\right)}^{2}\times 5000})}=500$$

*n* = population size (9000); e = margin of error (percentage in decimal form); Z = 1.96 (95% Confidence Interval); prevalence of substance use = 50%.

The calclated minimum sample size is 500. However, for this study, 791 students (*n* = 791) were included to enhance the generalizability and statistical power of the study.

### 2.3. Data Collection and Measures

The study questionnaire included sociodemographic variables such as age, gender, education level, family income, parental education, and family history of substance use.

Psychoactive substances (PASs) encompassed tobacco, alcohol, cannabis, cocaine, heroin, ecstasy, and psychotropic medications used without a medical prescription. Participants were asked to indicate whether they had used any of these substances at least once in their lifetime, within the past 12 months, in the last month, and whether they were current users.

The Diagnostic and Statistical Manual of Mental Disorders (DSM-5) criteria for substance use disorders (SUDs) were used to assess the severity of substance-related disorders, categorized as follows: no SUD (<2 criteria met), mild SUD (2–3 criteria), moderate SUD (4–5 criteria), and severe SUD (>6 criteria) [24].

The Ask Suicide-Screening Questions (ASQ) tool was used to assess suicide risk. It consists of five items with binary response options (“yes” or “no”). If an individual answered “no” to all of questions 1 to 4, the screening was concluded and considered negative. A positive screening was indicated if the response to any of these four questions was “yes.” In such cases, question 5 was then asked to determine the immediacy of the risk. A “yes” response to question 5 indicated an acute positive screening, signifying imminent risk. Conversely, a “no” response to question 5 indicated a non-acute positive screening, suggesting potential but not immediate risk. This approach allowed for differentiation between immediate and potential suicide risk [25].

To assess the frequency of suicidal ideation among study participants, the Suicidal Ideation Attributes Scale (SIDAS) was employed, comprising five items. Participants provided responses on a 10-point scale. The first question addressed the frequency of suicidal thoughts in the past month, ranging from 0 (never had this thought) to 10 (always). The second question evaluated the degree of control over these thoughts, with 0 indicating no control and 10 indicating total control. The third question assessed the proximity of a suicide attempt in the past month, ranging from 0 (not at all) to 10 (attempted). The fourth question assessed the degree of distress caused by suicidal thoughts, ranging from 0 (not at all) to 10 (extremely). Lastly, the fifth question evaluated the extent to which suicidal thoughts interfered with daily activities, such as work, household tasks, or social activities, with 0 indicating no interference and 10 indicating extreme interference. This scale facilitated the measurement of different dimensions of suicidal ideation and the quantification of its impact on the participants’ daily lives [26].

The assessment of early trauma involved using the Adverse Childhood Experiences (ACE) questionnaire, which comprises 10 items, with responses in the form of yes or no. Participants received a score based on their exposure to adverse experiences. A score of 0 to 1 indicated relatively low exposure to such experiences, while a score of 2 to 3 denoted moderate exposure. A score of 4 to 5 reflected moderate to high exposure, and a score of 6 to 7 indicated high exposure to negative experiences. Finally, a score of 8 to 10 represented very high exposure to adverse experiences [27].

The presence of depressive symptoms among participants was assessed according to the diagnostic criteria for depression outlined in DSM-5. This tool comprised nine questions regarding depressive symptoms, such as markedly decreased interest or pleasure in almost all activities for nearly every day. Participants responded with yes or no to each question. To meet the diagnostic criteria for depression, participants needed to exhibit at least five of these symptoms for a continuous period of two weeks, accompanied by a deviation from their usual functioning. Scores obtained were categorized based on the number of symptoms present: from 0 to 4 for no depression, 5 to 9 for mild depression, 10 to 14 for moderate depression, 15 to 19 for moderately severe depression, and 20 or more for severe depression [28]. The scales employed in this study have been specifically adapted and validated for utilization among adolescents, as evidenced by prior studies [29,30,31].

### 2.4. Ethical Considerations and Procedure

Informed consent was acquired via telephone from the parents, while the participants themselves provided consent in person prior to their involvement in the study. The participants were informed of the study’s objectives, procedures, and the confidentiality maintained throughout the research. The regional academy of education and teaching granted authorization for the study, along with the ethical committee’s approval (REF: FCR-CS-07/2023-0001). Verbal informed consent was acquired from the participants, and instructions and clarifications were provided immediately whenever necessary. Confidentiality and the protection of privacy for the gathered data were maintained by employing methods of anonymous data collection.

### 2.5. Statistical Analyses

The data were initially input into Microsoft Excel for coding and filtering, and were subsequently analyzed using SPSS v.25.0 software. Categorical data were summarized in tables displaying frequencies and percentages. Continuous variables were presented as means and standard deviations (±). Univariate logistic regression was employed to explore associations between suicidal ideation and various independent variables, including sociodemographic factors (age, gender, education level, family income, and parental education), substance use behaviors (tobacco, alcohol, cannabis, and other psychoactive substances), mental health indicators (suicidal risk assessed by ASQ and depressive symptoms based on DSM-5 criteria), and early trauma exposure (ACE questionnaire scores). Each variable was analyzed individually to determine its crude association with suicidal ideation before being included in the multivariate analysis to control for potential confounders. All associations with a *p*-value < 0.05 in the univariate analysis were further examined in the multivariate analysis to identify distinct associations with suicidal ideation and mitigate confounding factors. Moreover, 95% confidence intervals were estimated for explanatory variables. A *p*-value < 0.05 was considered statistically significant.

For Pearson correlation, the correlation coefficient (*r*) served to measure both the strength and direction of an association. Ranging from −1 to +1, it equals 0 in the absence of any association. The closer this coefficient is to −1 or +1, the stronger the association between the two variables. Its threshold of significance is as follows: ns, non-significant: >1 or <1 or =0; *, low correlation: <0.5 or >−0.5; **, moderate correlation: >0.5 or <−0.5; ***, high correlation: =1 or =−1 [32].

## 3. Results

A total of 791 students participated in the study. The average age of the participants was 15.59 years, with ages ranging from 11 to 23 years (standard deviation = 2.15). In terms of gender distribution, 52% of the participants were male, while 48% were female. Regarding family structure, 98% of participants lived with their biological parents on a daily basis, with 1.9% being adopted. In terms of residence, 79% of participants lived in urban areas, while 21% resided in rural areas. Regarding the employment status of parents, 91% of fathers were employed, while 85% of mothers were not working. As for the educational level of the participants, 11.6% were in their first year of middle school, 13.3% in their second year of middle school, 16.3% in their third year of middle school, 16.3% in their first year of high school, 15% in their second year of high school, and 26.8% in their third year of high school. These data provide an overview of the demographic characteristics of the study participants (Table 1).

Binary logistic regression analysis revealed several significant associations with suicidal ideation. Female individuals exhibited a higher risk of suicidal ideation compared to male individuals (OR 1.16 [1.1–2.34]; *p* < 0.01). Conversely, those living with their biological parents had a lower risk of suicidal ideation (OR 0.29 [0.1–0.8]; *p* < 0.01). Regarding medical history, those whose parents had such history were at a higher risk (OR 1.67 [1.12–2.5]; *p* < 0.01), as well as those whose parents had psychiatric histories, with an elevated risk for both mothers (OR 1.95 [1.02–3.72]) and fathers (OR 2.91 [1.19–7.09]). Concerning parental substance use, those with using parents had a higher risk (OR 6.7 [1.48–30.32] for mothers and OR 2.76 [1.87–4.08] for fathers). These findings underscore the significance of family factors and medical/psychiatric histories in suicidal ideation among the study participants (Table 1).

Regarding the participants’ relationships with their parents, approximately 65% reported having good to very good social relationships, while 32% described their relationships as moderate. Only 2.9% of participants indicated having poor relationships with their parents. In terms of relationships with siblings, the majority (65.7%) reported poor relationships. As for friendships and interactions with teachers, 45.3% of participants maintained good relationships with their friends, and 47.5% reported good relationships with their teachers (Table 2).

The logistic regression analysis showed that participants with poor relationships in their environment were at a higher risk of suicidal ideation. Specifically, participants with poor relationships with their parents had an increased risk, with an OR of 5.22 (95% CI: 2.11–12.86). Similarly, those with poor relationships with their friends had an increased risk, with an OR of 2.41 (95% CI: 1.01–5.72), and those with poor relationships with their teachers had an increased risk, with an OR of 3.88 (95% CI: 1.84–8.16). These findings highlight the significance of positive social relationships with parents, friends, and teachers in preventing suicidal ideation among participants (Table 2).

A statistically significant difference in sleep patterns was observed (χ^2^ = 64; *p* < 0.001) between participants with suicidal ideation and others. Notably, 23.9% of those with suicidal ideation reported sleeping less than 4 h, compared to just 6% of participants without suicidal ideation. Regarding personal history, several factors were found to increase the risk of suicidal ideation. Participants with a medical history exhibited a significantly higher risk, with an odds ratio (OR) of 2.48 (95% CI: 1.62–3.81; *p* < 0.001). Those with a history of surgery also had an increased risk, with an OR of 1.85 (95% CI: 1.12–3.08; *p* < 0.001). Furthermore, participants with a legal history showed an elevated risk, with an OR of 2.84 (95% CI: 1.33–6.09; *p* = 0.007) (Table 3).

The current study found that the prevalence of suicidal ideation among participants was 16.94%. Logistic regression analysis revealed significant associations between suicidal ideation and early trauma, as indicated by the ACE score (OR 4.72 [95% CI 2.49–8.94]; *p* < 0.001), depression (OR 11.47 [95% CI 6.94–18.96]; *p* < 0.001), and substance abuse (OR 2.10 [95% CI 1.19–3.7]; *p* < 0.001) (Table 4).

The findings of this study indicate that 25% of participants are currently using psychoactive substances (Figure 1).

Regarding the reasons for use, curiosity was the primary reason for 16.7% of participants and the current reason for 4%. However, the use of psychoactive substances (PASs) to forget problems has increased from 2.4% at initial use to 40% at present. Similarly, the use of substances for pleasure has risen from 3.3% to 40% (Figure 2).

Regarding the types of substances used, tobacco was the most frequently used substance initially, accounting for 18.6% of cases, closely followed by MDMA at 2.66%. For current use, tobacco remained dominant at 11.3%, followed by MDMA at 5%, cannabis at 1.8%, and alcohol at 0.8%. In terms of associated substances, alcohol accounted for 2.3% of cases and psychotropic medications for 1.1% (Figure 3).

According to the data from Table 5, the average age at onset for psychoactive substance use is 14.28 years, with a standard deviation of 1.7. The minimum age at onset for substance use is 7 years, while the maximum age is 14.2 years. Regarding monthly expenses dedicated to psychoactive substances (PASs), the average is 454.83, with a standard deviation of 480.76.

Pearson correlation analysis revealed several significant associations among the measured variables. Age was positively correlated with suicidal risk (*r* = 0.109; *p* = 0.002), expenses (*r* = 0.195; *p* < 0.001), depression (*r* = 0.182; *p* < 0.001), and substance use (*r* = 0.172; *p* < 0.001), indicating that as age increases, these variables tend to rise too. Expenses also showed positive correlations with depression (*r* = 0.119; *p* < 0.001), suicidal risk (*r* = 0.145; *p* < 0.001), and substance use (*r* = 0.138; *p* < 0.001).

Early trauma was positively correlated with all the measured variables, with correlation coefficients ranging from 0.043 to 0.477 (*p* < 0.001). Similarly, depression had positive correlations with all the variables, with coefficients ranging from 0.193 to 0.552 (*p* < 0.001). Suicidal risk also demonstrated positive correlations with all measured variables, with coefficients ranging from 0.217 to 0.552 (*p* < 0.001). Furthermore, suicidal ideation at birth and in the last month were both positively correlated with all variables, with coefficients ranging from 0.236 to 0.777 (*p* < 0.001). Finally, substance use showed positive correlations with suicidal risk (*r* = 0.090; *p* = 0.018), suicidal ideation at birth (*r* = 0.044; *p* < 0.001), and the ASQ score (*r* = 0.136; *p* < 0.001) (Table 6).

## 4. Discussion

The study investigated the prevalence of suicidal ideation, psychoactive substance use, and certain psychiatric disorders among middle and high school students in Marrakech. The results revealed a prevalence of 16.94% for suicidal ideation and 25% for current psychoactive substance use among participants. The primary reasons for use were curiosity (16.7% for initial use and 4% for current use) and using substances to cope with problems (ranging from 2.4% to 40% over time). Tobacco was the most commonly used substance, both for initial use (18.6%) and current use (11.3%). Correlations demonstrated significant associations between suicidal ideation and early trauma, depression, and substance use. Moreover, age was positively correlated with suicide risk, expenditures, depression, and expenses on substances.

This study found that one in six students reported suicidal ideation. These results are consistent with prior studies. In a study conducted by Tom and Mahfoud [19], a similar trend was observed, where approximately 16.66% of school-going adolescents in Morocco acknowledged experiencing suicidal ideation, and around 14.28% reported making plans related to suicide. Another study conducted in northern Morocco found a prevalence of 15% in students aged 13 to 15 [33]. The current results revealed that females exhibited a higher risk of suicidal ideation compared to males (OR 1.16 [1.1–2.34]; *p* < 0.01), which contrasts with earlier research findings, where the prevalence rates of suicide attempts were often reported as two to three times higher in females than in males [34].

The findings underscore a significant relationship between suicidal ideation and psychoactive substance (PAS) use. These results suggest a bidirectional association, where regular PAS use increases the risk of suicidal ideation, and suicidal ideation can, in turn, amplify PAS use. This finding concurs with the conclusions of earlier studies, such as that conducted by Zhang and Wu [35], which highlighted a significant correlation between smoking/alcohol consumption and suicidal ideation, but no association between drug use (such as marijuana) and suicidal ideation. Nonetheless, previous studies have also emphasized a positive correlation between tobacco use, PAS use, and suicidal behaviors [36,37,38].

The connection between substance use disorder and suicide implies that individuals might develop substance dependence as a substitute for suicidal tendencies. For example, individuals might turn to consuming harmful substances to divert themselves from thoughts of self-harm. Moreover, an alternative explanation proposes that suicide could be a result of adverse experiences linked with substance use disorder, including social isolation, stigma, and discrimination [39].

Janssen et al. (2017) examined the impact of cannabis consumption on suicidal ideation among 491 school adolescents. They revealed an association between suicidal ideation, depressive symptoms, and cannabis use during adolescence [40]. Cannabis consumption emerged as an independent predictive factor for suicidal ideation, even after controlling for depressive symptoms. Nevertheless, depressive symptoms were found to be the primary predictor of suicidal ideation, while cannabis consumption contributed only marginally [41]. The relationship between substance-related disorders and suicide suggests that substance dependence can develop as an alternative to suicide. For instance, individuals may turn to consuming harmful substances to alleviate suicidal thoughts. Furthermore, another explanatory model posits that suicide results from negative experiences accompanying substance-related disorders, such as social exclusion, stigma, and discrimination [39].

Depression and anxiety represent critical mental health challenges that significantly impact the lives of adolescents, potentially giving rise to family conflicts, substance misuse, social difficulties, violence, and even suicide [42]. The findings of this study underscore a rise in contemporary youth experiencing suicidal ideation, and these thoughts are positively linked with depression among participants. Furthermore, this study identifies depression as a predictor of suicidal ideation. Previous research similarly identified depression as a prime factor of suicidal ideation in young individuals, as evidenced by the majority of adolescents who had attempted suicide meeting diagnostic criteria for depression [13,43]. Notably, such suicidal ideation heightened the vulnerability to psychiatric issues, suicide attempts, and completed suicides [44,45]. An earlier investigation revealed that by the age of 18, approximately 11% of youth suffered from depressive disorders, with contributory risk factors encompassing substance abuse, self-esteem challenges, relationship difficulties, family history, psychological concerns, and mental health conditions [46]. Despite the association between depression and suicidal ideation, Pinto and Whisman (1996) suggested that the presence of depressive symptoms alone might not be adequate for accurately predicting the risk of suicide [47].

The high prevalence of exposure to various types of early-life stress highlights the importance of implementing therapeutic initiatives and practices aimed at mitigating the consequences of such exposure. The group of participants who had suicidal ideation displayed greater exposure to different forms of abuse compared to those who had never attempted suicide, particularly in terms of emotional and physical abuse. This finding aligns with prior studies that have shown a link between early-life stress exposure and suicidal behaviors. Additionally, individuals diagnosed with substance-related disorders who had attempted suicide exhibited a prevalence of exposure to childhood traumas [48,49]. A biological explanation for the relationship between childhood traumas and suicide refers to epigenetic modifications caused by early-life stress, stemming from the hyperactivity of the hypothalamic–pituitary–adrenal axis and an heightened stress response in patients exposed to childhood traumas [50].

Furthermore, the outcomes of this study demonstrated that age was positively correlated with suicidal ideation. This finding is in contrast to some prior studies, such as the one by Fergusson et al., who observed that the prevalence of lifetime suicidal ideation increased from 9.5% at age 16 to 24.5% at age 21. However, our study found that age was a significant factor, with a positive correlation suggesting that older adolescents are at a higher risk of suicidal ideation [48]. According to Ibrahim et al., the highest rates of ideation and plans in the past year were during mid-adolescence for girls, whereas these rates slowly increased through late adolescence in boys. This pattern can be attributed to their comparatively limited coping skills and life experiences, as well as their problem-solving approaches [51]. Generally, women tend to address problems by seeking assistance from those close to them, engaging in detailed discussions about the situation and potential solutions. Conversely, men often adopt problem-solving methods that involve less communication, potentially causing them to internalize issues, which in turn could lead to suicidal ideation. This divergence might reflect disparities in developmental maturity between genders, with women typically maturing earlier than men. This developmental gap could indirectly contribute to the contrast in coping abilities and problem-solving strategies between men and women [52,53].

This study contributes to the scarce data existing in the region. Drawing from the outcomes and considering regional patterns, there is a need to examine national systemic measures aimed at reducing the increasing public health concern. The interventions should target the reduction in hunger occurrence, creating supportive school environments, the incorporation of school counselors to identify initial signs and implement timely interventions, and the provision of education on mental health alongside the consequences of cigarette smoking, alcohol, and substance abuse. Moreover, significant efforts are needed to address the stigma associated with mental illness, as it influences individuals’ readiness to seek assistance [19].

This study has several limitations. In terms of its research design, a cross-sectional approach was employed and thus no causal relationship is established. Since all measurements were captured at a singular time point, the sequencing of variables over time could not be ascertained. Moreover, the inclusion of self-assessments introduces the potential for social desirability bias. The consideration of non-response bias is essential. Typically, individuals who take part in health surveys exhibit better health compared to those who choose not to participate. To gain a more comprehensive understanding of the intricate mechanisms underpinning suicidal ideation and the usage of psychoactive substances among adolescents and young adults, it becomes imperative to conduct longitudinal investigations and more extensive clinical assessments.

## 5. Conclusions

Suicide is a complex and multifactorial issue with devastating consequences for individuals, families, and communities. The findings of this study underscore the importance of considering multiple risk factors in the assessment and prevention of suicide, such as a history of trauma, depression, anxiety, and psychoactive substance use. The connections between these factors often exhibit bidirectional and significant associations. It is crucial to implement strategies for the early screening, management, and treatment of individuals exhibiting suicidal ideation and mental disorders in order to alleviate the burden of suicide. Additionally, preventive and awareness initiatives need to be established to promote mental health, reduce stigma, and provide social support for individuals facing psychological distress. Mental health screenings in schools and communities are essential for early identification and intervention. Programs should focus on training educators and health professionals to recognize early signs of suicidal ideation and substance abuse. Community awareness campaigns are needed to reduce stigma and promote mental health support, while implementing trauma-informed care to mitigate the impact of early life stress. Policymakers should prioritize access to mental health resources and crisis intervention programs, ensuring that individuals in distress receive timely and appropriate care.

## Figures and Tables

**Figure 1 healthcare-13-01178-f001:**
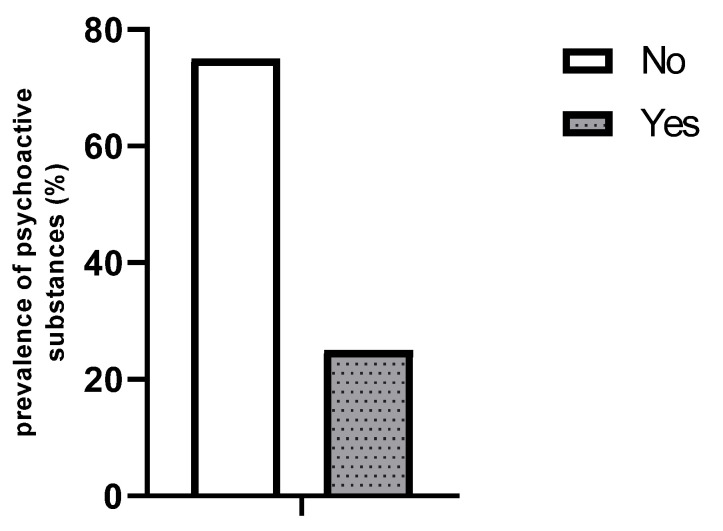
Prevalence of psychoactive substance use among participants.

**Figure 2 healthcare-13-01178-f002:**
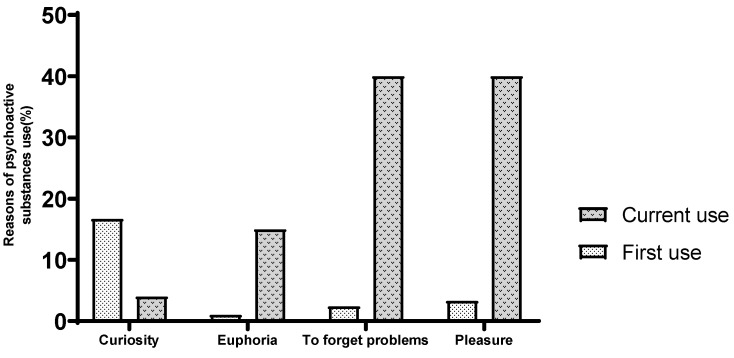
Prevalence of PASs among participants.

**Figure 3 healthcare-13-01178-f003:**
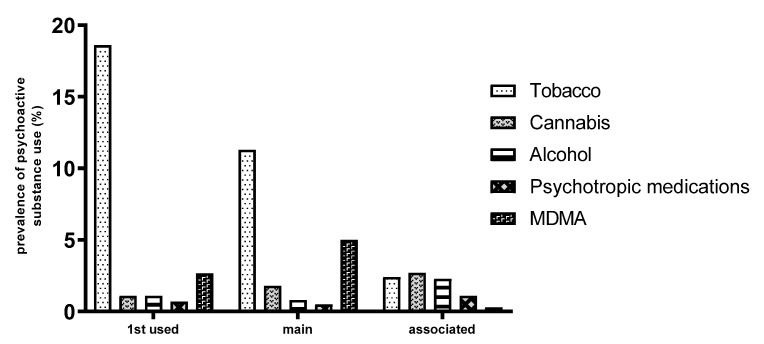
Prevalence of initial, primary, and associated substance use among participants.

**Table 1 healthcare-13-01178-t001:** Suicidal ideation and sociodemographic characteristics of participants.

		Suicidal Ideation					
Variable	Modality	No (*n* = 657)	Yes (*n* = 134)	χ^2^	*p*	Odds Ratio (OR) (IC 95%)	*p*
Age	Mean (±ET)	15.98 (2.17)	15.98 (2.07)	-	-	-	-
Gender	Female	300 (45.7)	77 (57.5%)	6.2	0.01	1.16 (1.1–2.34)	0.01
	Male	357 (54.3)	57 (42.5%)			1	
Parents	Biological	648 (98.6%)	127 (94.8%)	10.74	0.005	0.29 (0.1–0.8)	0.02
	Adoptive	9 (1.4%)	6 (4.5%)			1	
Educational Level	1st M. School	74 (11.3%)	18 (13.4%)	Ns	Ns	Ns	Ns
	2nd M. School	92 (14.0%)	18 (13.4%)				
	3rd M. School	109 (16.6%)	19 (14.2%)				
	1st H. School	105 (16.0%)	24 (17.9%)				
	2nd H. School	102 (15.5%)	18 (13.4%)				
	3rd H. School	175 (26.6%)	37 (27.6%)				
Place of Residence	Rural	154 (23.4%)	14 (10.4%)	12.4	0.006	Ns	
	Urban	503 (76.6%)	120 (89.6%)				
Father’s Working Status	No	55 (8.4%)	11 (8.2%)	Ns	Ns	Ns	ns
	Yes	602 (91.6%)	123 (91.8%)				
Mother’s Working Status	No	562 (85.5%)	113 (84.3%)	Ns	Ns	Ns	ns
	Yes	95 (14.5%)	95 (14.5%)				
Mother’s Medical History	No	501 (76.3%)	88 (65.7%)	6.55	0.01	1	0.01
	Yes	156 (23.7%)	46 (34.3%)			1.67 (1.12–2.5)	
Father’s Medical History	No	536 (81.6%)	101 (75.4%)	Ns	Ns	Ns	ns
	Yes	121 (18.4%)	33 (24.6%)				
Mother’s Psychiatric History	No	620 (94.4%)	120 (89.6%)	4.28	0.003	1	0.04
	Yes	37 (5.6%)	14 (10.4%)			1.95 (1.02–3.72)	
Father’s Psychiatric History	No	643 (97.9%)	126 (94.0%)	6.01	0.01	1	0.01
	Yes	14 (2.1%)	8 (6.0%)			2.91 (1.19–7.09)	
Father’s Legal History	No	641 (97.6%)	122 (91.0%)	13.85	<0.001	1	0.001
	Yes	16 (2.4%)	12 (9.0%)			3.94 (1.81–8.53)	
Mother’s Substance Use	No	654 (99.5%)	130 (97.0%)	8.12	0.004	1	0.01
	Yes	3 (0.5%)	4 (3.0%)			6.7 (1.48–30.32)	
Father’s Substance Use	No	515 (78.4%)	76 (56.7%)	27.66	<0.001	1	<0.001
	Yes	142 (21.6%)	58 (43.3%)			2.76 (1.87–4.08)	

**Table 2 healthcare-13-01178-t002:** Suicidal ideation and relationships with the environment.

		Suicidal Ideation					
Variable	Modality	No (*n* = 657)	Yes (*n* = 134)	χ^2^	*p*	OR (IC 95%)	*p*
Parental Relationships	Very Good	224 (34.1%)	33 (24.6%)	28.68	<0.001	1	
	Good	227 (34.6%)	30 (22.4%)			ns	
	Average	193 (29.4%)	61 (45.5%)			2.14 (1.34–3.41)	0.001
	Poor	13 (2.0%)	10 (7.5%)			5.22 (2.11–12.86	<0.001
Sibling Relationships	Very Good	204 (31.4%)	28 (20.9%)	33.82	<0.001	0.37 (0.16–0.84)	0.01
	Good	316 (48.1%)	47 (35.1%)			0.4 (0.18–0.88)	0.02
	Average	108 (16.4%)	49 (36.6%)			ns	
	Poor	27 (4.1%)	10 (7.5%)			1	
Friend Relationships	Very Good	183 (27.9%)	31 (23.1%)	13.36	0.004	1	
	Good	309 (47.0%)	49 (36.6%)			ns	
	Average	143 (21.8%)	45 (33.6%)			1.85 (1.11–3.08)	0.01
	Poor	22 (3.3%)	9 (6.7%)			2.41 (1.01–5.72)	0.04
Teacher Relationships	Very Good	138 (21.0%)	20 (14.9%)	15.25	0.002	1	
	Good	316 (48.1%)	60 (44.8%)			ns	
	Average	171 (26.0%)	36 (26.9%)			ns	
	Poor	32 (4.9%)	18 (13.4%)			3.88 (1.84–8.16)	<0.001

**Table 3 healthcare-13-01178-t003:** Suicidal ideation and participants’ background.

		Suicidal Ideation					
Variable	Modality	No	Yes	χ^2^	*p*	OR (IC 95%)	*p*
Sleep duration	>10 h	71 (10.8%)	30 (22.4%)	64.39	<0.001	ns	
	between 8 h and 10 h	171 (26.0%)	29 (21.6%)				
	between 6 h and 8 h	375 (57.1%)	43 (32.1%)				
	<4 h	40 (6.1%)	32 (23.9%)				
Medical history	No	561 (85.4%)	94 (70.1%)	18.15	<0.001	1	<0.001
	Yes	96 (14.6%)	40 (29.9%)			2.48 (1.62–3.81)	
Surgical history	No	588 (89.5%)	110 (82.1%)	5.88	0.01	1	0.001
	Yes	69 (10.5%)	24 (17.9%)			1.85 (1.12–3.08)	
Legal history	No	637 (97.0%)	123 (91.8%)	7.88	0.005	1	0.007
	Yes	20 (3.0%)	11 (8.2%)			2.84 (1.33–6.09)	

**Table 4 healthcare-13-01178-t004:** Suicidal ideation and participants’ psychiatric history.

		Suicidal Ideation					
Variable	Modality	No	Yes	χ^2^	*p*	OR (IC 95%)	*p*
Early trauma	No	195 (29.7%)	11 (8.2%)	26.46	<0.001	1	<0.001
	Yes	462 (70.3%)	123 (91.8%)			4.72 (2.49–8.94)	
DSM depression	No	439 (66.8%)	20 (14.9%)	123	<0.001	1	<0.001
	Yes	218 (33.2%)	114 (85.1%)			11.47 (6.94–18.96)	
DSM addiction	No	588 (89.5%)	102 (76.1%)	22.00	<0.001	1	
	Mild	4 (0.6%)	3 (2.2%)			Ns	
	Moderate	13 (2%)	10 (7.5%)			4.43 (1.89–10.38)	<0.001
	Severe	52 (7.9%)	19 (14.2%)			2.10 (1.19–3.7)	<0.001

**Table 5 healthcare-13-01178-t005:** Age at onset and consumption expenses of psychoactive substance users.

	Age at Onset	PAS Consumption Expenses (MAD)
Minimum	7	20
Maximum	19	2500
Mean	14.28	454.83
Standard Deviation (*n*)	1.70	480

**Table 6 healthcare-13-01178-t006:** Correlation between depression, psychoactive substance use, and suicidal ideation in participants.

		1	2	3	4	5	6	7	8	9
Age (1)	*r*	1	0.195 **	0.052	0.152 **	0.119 **	−0.054	0.025	0.003	0.172 **
	*p*		0.000	0.170	0.000	0.002	0.158	0.509	0.942	0.000
Expenses (2)	*r*	0.109 **	1	0.068	0.119 **	0.145 **	0.029	0.121 **	0.132 **	0.651 **
	*p*	0.002		0.104	0.005	0.001	0.493	0.004	0.002	0.000
ACE score (3)	*r*	0.043	0.068	1	0.441 **	0.477 **	0.126 **	0.161 **	0.215 **	0.045
	*p*	0.225	0.104		0.000	0.000	0.001	0.000	0.000	0.237
DSM depression (4)	*r*	0.182 **	0.119 **	0.441 **	1	0.503 **	0.193 **	0.552 **	0.523 **	0.077 *
	*p*	0.000	0.005	0.000		0.000	0.000	0.000	0.000	0.043
Suicidal risks (5)	*r*	0.109 **	0.145 **	0.477 **	0.503 **	1	0.217 **	0.313 **	0.372 **	0.090 *
	*p*	0.002	0.001	0.000	0.000		0.000	0.000	0.000	0.018
Suicidal ideation at birth (6)	*r*	0.011	0.029	0.126 **	0.193 **	0.217 **	1	0.236 **	0.300 **	0.044
	*p*	0.754	0.493	0.001	0.000	0.000		0.000	0.000	0.252
Suicidal ideation last month (7)	*r*	0.048	0.121 **	0.161 **	0.552 **	0.313 **	0.236 **	1	0.732 **	0.127 **
	*p*	0.176	0.004	0.000	0.000	0.000	0.000		0.000	0.001
SIADS (8)	*r*	0.016	0.132 **	0.215 **	0.523 **	0.372 **	0.300 **	0.732 **	1	0.126 **
	*p*	0.643	0.002	0.000	0.000	0.000	0.000	0.000		0.001
DSM substance abuse (9)		0.173 **	0.651 **	0.045	0.077 *	0.090 *	0.044	0.127 **	0.126 **	1
		0.000	0.000	0.237	0.043	0.018	0.252	0.001	0.001	

* low correlation; ** moderate correlation.

## Data Availability

Data are contained within the article.

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
