# Peer review of "Suicidal Ideation and Substance Use Among Middle and High School Students in Morocco"

_healthcare, 2025, doi:10.3390/healthcare13101178_

Round 1
Reviewer 1 Report
Comments and Suggestions for Authors
- In line 37, please capitalize the "s" in suicide. Also, rephrase to the following: "Suicide accounts for 1.4% of global mortality, positioning itself as the 15th leading cause of death worldwide, and as the second primary cause of death among individuals aged 15 to 29."
- In line 44, the comma should be replaced with a period.
- In line 56, replace "and the like", with "among others".
- In line 58, remove "the" before "youth"
- In line 72, add "underreporting of cases"
- I believe the introduction lacks the following:
- An overview of studies showing the association between suicidal ideation and substance use in youth in the Middle East to further highlight the gap in research.
- More studies showing the association between suicidal ideation and substance use specifically rather than focusing on broader mental health issues.
- In line 92-93, can you elaborate on the standardized questionnaire used?
- In line 113, add that the sample size was increased to improve generalizability and power of the study.
- In line 183, please elaborate on the associations explored in univariate analysis.
- The second paragraph in the discussion contradicts what is reported in the results in terms of having males at higher risk of reporting suicidal ideation rather than no gender-significant differences. Please clarify.
- In line 377, replace "didn't" with "did not".
- In table 6, please define what *** mean
- the discussion also contradicts the information that age is not a predictor of suicidal ideation although the results show a positive correlation. Please clarify.
- Please make sure to address grammatical errors throughout the manuscript.
Grammatical errors and language issues were detected across manuscript.
Author Response
Q.1In line 37, please capitalize the "s" in suicide. Also, rephrase to the following: "Suicide accounts for 1.4% of global mortality, positioning itself as the 15th leading cause of death worldwide, and as the second primary cause of death among individuals aged 15 to 29.
Response 1: Thank you for your comment. The sentence has been corrected to: “Suicide represents 1.4% of global deaths, making it the 15th leading cause of mortality worldwide and the second leading cause of death among individuals aged 15 to 29.
Q2.In line 44, the comma should be replaced with a period.
Response 2: Thank you for your comment. The punctuation has been corrected.
Q3.In line 56, replace "and the like", with "among others".
Response 3: Thank you for your comment. The phrase has been corrected as suggested.
Q4. In line 58, remove "the" before "youth"
Response 4: Thank you for your comment. "The" before "youth" has been removed.
Q5.In line 72, add "underreporting of cases"
Response 5: Thank you for your comment. The phrase "underreporting of cases" has been added.
Q6.I believe the introduction lacks the following:
Q6.1An overview of studies showing the association between suicidal ideation and substance use in youth in the Middle East to further highlight the gap in research.
Response 6.1: A new paragraph has been added, discussing the research gap on suicidal ideation and substance use among youth in the Middle East and North Africa, particularly in Morocco. This paragraph also highlights cultural and social barriers such as underreporting and stigma, which further justify the need for this study
Q6.2More studies showing the association between suicidal ideation and substance use specifically rather than focusing on broader mental health issues.
Response 6.2: The revised version explicitly addresses the connection between substance use (e.g., alcohol, tobacco) and suicidal ideation, emphasizing that much of the existing literature tends to focus on broader mental health issues rather than this specific link.
Q7.In line 92-93, can you elaborate on the standardized questionnaire used?
Response 7: Thank you for your comment. Additional details regarding the structured questionnaire have been incorporated.
Q8. In line 113, add that the sample size was increased to improve generalizability and power of the study.
Response 8: Thank you for your comment. A sentence has been added to specify that the sample size was increased to improve the generalizability and statistical power of the study.
Q9.In line 183, please elaborate on the associations explored in univariate analysis.
Response 9: Thank you for your comment. The associations explored in the univariate analysis have been elaborated upon.
Q10.The second paragraph in the discussion contradicts what is reported in the results in terms of having males at higher risk of reporting suicidal ideation rather than no gender-significant differences. Please clarify.
Response 10: Thank you for your valuable feedback. We acknowledge the inconsistency in the discussion regarding gender differences in suicidal ideation. Upon review, we confirmed that female individuals exhibited a higher risk of suicidal ideation compared to males (OR 1.16 [1.1-2.34], p < 0.01). The discussion has been revised accordingly to align with the results and relevant literature, emphasizing that females are at higher risk.
Q11. In line 377, replace "didn't" with "did not".
Response 11: Thank you for your comment. "Didn't" has been replaced with "did not."
Q12. In table 6, please define what *** mean
Response 12: Thank you for your comment. A note has been added at the end of Table 6 to define "***" as indicating statistical significance.
Q13. the discussion also contradicts the information that age is not a predictor of suicidal ideation although the results show a positive correlation. Please clarify.
Response 13: Thank you for your feedback. The discussion has been revised to clarify that our results show a positive correlation between age and suicidal ideation, with older adolescents being at a higher risk. This aligns with previous findings, such as those from Fergusson et al., who reported an increase in lifetime suicidal ideation prevalence with age. The revised discussion ensures consistency with our results.
Q14. Please make sure to address grammatical errors throughout the manuscript.
Response 14: Thank you for your comment. The entire manuscript has been thoroughly reviewed, and grammatical errors have been corrected.
Reviewer 2 Report
Comments and Suggestions for Authors
The authors presented the results of a study on the frequency of suicidal ideation and its associations with substance use, depression, and traumatic childhood experiences. They also presented the sociometric characteristics of the study group.
The topic is important and has recently attracted the attention of many researchers (psychologists, sociologists, doctors, politicians, teachers, and generally - the general public) due to the reported increase in the frequency of recognizing symptoms of depression in the world - and what is very worrying - in many young people.
The study used data collection tools, namely the Suicidal Ideation Attributes Scale (SIDAS), the Diagnostic and Statistical Manual of Mental Disorders (for substance use and depression), and the Adverse Childhood Experiences (ACE) (for recognizing traumatic childhood experiences).
The study involved a group of 791 young people in adolescence and transition to early adulthood (aged 11 to 23) - all of them attended schools in central Morocco.
Stratified random sampling was used. The criteria for inclusion in the study group were defined.
The discussion that was conducted is interesting. The focus was on the most important issues from the point of view of the purpose of the research. In explaining the results of the research on the "destructive" impact of early trauma, the authors referred to findings from biology. They signaled certain discrepancies in the results of the research on the differences in the frequency of suicide attempts between girls/women and boys/men. In their explanation, the authors refer, among other things, to the earlier biopsychosocial maturation of the female sex. It would be worth paying more attention to the age of the subjects (in the compared studies), the specificity of the cultural context, including family socialization. Attention was drawn to the multiplicity of risk factors for the frequency of suicidal thoughts, use of psychoactive substances, early childhood trauma, depressive symptoms and their mutual connections. The importance of selected sociodemographic factors was taken into account as potentially significant for the above relationships. It would be interesting to present the specific cultural characteristics of Morocco and to outline the specifics of the current socio-economic situation in the country, which may be important for the sense of well-being or hopelessness of the young generation.
The points in the discussion are interesting, including the possibility of considering substance addiction as a substitute for suicidal tendencies (worth considering in more detail – not necessarily in this article but in another study). I wonder if substance addiction can be treated as a suicide spread out/stretched over time? As a continuous, specific self-harm? (unconscious tendency)?
Referring to one of the limitations of the study noticed by the authors: What was the percentage of refusals to participate in the study or resignations from continuing the answer? If it is high, then one cannot rule out the assumption that the condition of young people in Morocco is worse than the obtained results indicate.
I wonder if getting consent from parents over the phone is a good enough solution.
I suggest you check the text carefully to remove a few inaccuracies. For example, "Another study conducted in the north of Morocco found a prevalence of 15% in students aged 13 to 15 % [30]." - “%” in line 319.
Overall, the study is very interesting and provides a basis for further exploration.
Author Response
The authors presented the results of a study on the frequency of suicidal ideation and its associations with substance use, depression, and traumatic childhood experiences. They also presented the sociometric characteristics of the study group. The topic is important and has recently attracted the attention of many researchers (psychologists, sociologists, doctors, politicians, teachers, and generally - the general public) due to the reported increase in the frequency of recognizing symptoms of depression in the world - and what is very worrying - in many young people. The study used data collection tools, namely the Suicidal Ideation Attributes Scale (SIDAS), the Diagnostic and Statistical Manual of Mental Disorders (for substance use and depression), and the Adverse Childhood Experiences (ACE) (for recognizing traumatic childhood experiences). The study involved a group of 791 young people in adolescence and transition to early adulthood (aged 11 to 23) - all of them attended schools in central Morocco. Stratified random sampling was used. The criteria for inclusion in the study group were defined. The discussion that was conducted is interesting. The focus was on the most important issues from the point of view of the purpose of the research. In explaining the results of the research on the "destructive" impact of early trauma, the authors referred to findings from biology. They signaled certain discrepancies in the results of the research on the differences in the frequency of suicide attempts between girls/women and boys/men. In their explanation, the authors refer, among other things, to the earlier biopsychosocial maturation of the female sex.
RESPONSE: Dear Reviewer, we sincerely appreciate your time and thoughtful evaluation of our manuscript. Below, we address each of your comments and outline the corresponding revisions made to improve the manuscript.
Q1.It would be worth paying more attention to the age of the subjects (in the compared studies), the specificity of the cultural context, including family socialization.
Response 1: Thank you for your comment. We have incorporated these aspects into the introduction.
Q2.It would be interesting to present the specific cultural characteristics of Morocco and to outline the specifics of the current socio-economic situation in the country, which may be important for the sense of well-being or hopelessness of the young generation.
Response 2: Thank you for your comment. These aspects have been added to the introduction.
Q3.The points in the discussion are interesting, including the possibility of considering substance addiction as a substitute for suicidal tendencies (worth considering in more detail – not necessarily in this article but in another study).
Response 3 : Thank you for your insightful comment. We acknowledge the importance of exploring the potential relationship between substance addiction and suicidal tendencies, particularly the idea that substance use may act as a substitute or coping mechanism for suicidal ideation in some individuals. This is indeed an interesting area that warrants further exploration. While this study focuses on the relationship between suicidal ideation and substance use in adolescents, we agree that the role of substance addiction as a possible alternative coping mechanism for suicidal tendencies is a promising direction for future research. As you suggest, this may not be fully explored in the current study, but it could form the basis of a more detailed investigation in future work. We believe this would offer valuable insights into how substance addiction may not only be a co-occurring risk factor but also a potential substitute for suicidal behaviors, particularly among young individuals grappling with emotional distress.
Q4. I wonder if substance addiction can be treated as a suicide spread out/stretched over time? As a continuous, specific self-harm? (unconscious tendency)?
Response 5 : Thank you for your insightful comment. We agree that substance addiction could be viewed as a long-term form of self-harm or chronic suicidal behavior, where individuals may unconsciously engage in substance use as a way to cope with emotional pain or hopelessness. While this idea falls outside the scope of the current study, we acknowledge its potential significance and suggest it could be explored in future research to better understand how substance addiction might reflect a gradual progression of suicidal tendencies.
Q5.Referring to one of the limitations of the study noticed by the authors: What was the percentage of refusals to participate in the study or resignations from continuing the answer? If it is high, then one cannot rule out the assumption that the condition of young people in Morocco is worse than the obtained results indicate.
Response 5 :Thank you for your comment. The percentage of refusals to participate or resignations from continuing the study was not high. Specifically, approximately 40 participants chose not to continue their involvement in the study. Given this small number, we believe the data collected still accurately reflects the situation among the majority of participants
Q6. I wonder if getting consent from parents over the phone is a good enough solution.
Response 6 : Thank you for your observation. In our study, we obtained parental consent over the phone, and we did not explicitly specify the exact focus of the research, framing it more generally as a study on mental health. While this approach was chosen for logistical reasons, we recognize that greater specificity in informing parents about the exact nature of the study (including the focus on suicidal ideation, substance use, and other sensitive topics) could have provided a clearer understanding of the research content. In hindsight, a more detailed explanation might have been beneficial, particularly considering the sensitive nature of the subject matter
Q7.I suggest you check the text carefully to remove a few inaccuracies. For example, "Another study conducted in the north of Morocco found a prevalence of 15% in students aged 13 to 15 % [30]." - “%” in line 319.
Reponse 7 ; Thank you for your comment. This specific issue has been corrected, along with other inaccuracies throughout the paper.
Round 2
Reviewer 1 Report
Comments and Suggestions for Authors
We thank the authors for addressing all comments. Kindly see below minor suggested changes:
- In line 107, indicate that the study is a cross-sectional study.
- Remove "is" in line 113 in "This is a cross-sectional study..." to avoid past and present tense issues.
Author Response
Q1In line 107, indicate that the study is a cross-sectional study.
R1. thank you dear reviewer it was added in the revised version
Q2. Remove "is" in line 113 in "This is a cross-sectional study..." to avoid past and present tense issues
R2. thank you, it is now corrected in the revised version
thank you for all your efforts
Round 3
Reviewer 1 Report
Comments and Suggestions for Authors
We thank the authors for addressing all my comments.
Author Response
We thank the authors for addressing all my comments.
response : Thank you so much for your time